# A $\delta^{11}B$ -pH calibration for the high-latitude foraminifera species Neogloboquadrina pachyderma and Neogloboquadrina incompta.

Elwyn de la Vega<sup>1\*</sup>, Markus Raitzsch<sup>2</sup>, Gavin L. Foster<sup>3</sup>, Jelle Bijma<sup>4</sup>, Ulyssess Ninnemann<sup>5</sup>, Michal Kucera<sup>6</sup>, Tali L. Babila<sup>7</sup>, Jessica Crumpton Banks<sup>1</sup>, Mohamed M. Ezat<sup>8</sup>, and Audrey Morley<sup>1,9\*</sup>.

Correspondence to: audrey.morley@universityofgalway.ie or elwyn.dlvega@gmail.com

Abstract. The boron isotopic composition of planktonic foraminifera is a powerful tool to reconstruct ocean pH 25 and CO<sub>2</sub> in the past. Applications to the high-latitude and polar oceans are however limited as robust calibrations between the  $\delta^{11}$ B of foraminifera and ocean pH in these regions are lacking. Here, we present a new empirical calibration for the high latitude Arctic species Neogloboquadrina pachyderma and the sub-polar to temperate species Neogloboquadrina incompta using towed specimens from the Labrador Sea, Baffin Bay, and the Nordic Seas. When paired with in situ hydrographic data, this approach allows us to avoid key assumptions used in 30 traditional core top calibrations that are required to link shell geochemical composition to hydrographic conditions during their formation. We show that the foraminifera  $\delta^{11}B$  of the species analysed is well correlated with the  $\delta^{11}B$ of seawater borate ion. Further, the foraminiferal  $\delta^{11}B$  values are consistently lower than seawater equilibrium borate values, consistent with the interpretation of more acidic seawater in the microenvironment due to 35 respiration. However, unlike published calibrations for non-spinose species to date the slope of the  $\delta^{11}B$ foraminifera to  $\delta^{11}B$  borate calibration is >1. We discuss several drivers of this higher sensitivity to pH and describe the possible role of vital effects in determining the boron isotopic composition of N. pachyderma and N.

<sup>&</sup>lt;sup>1</sup> University of Galway, Ryan Institute, and School of Geography, Archaeology and Irish Studies, H91TK33 Galway, Ireland.

<sup>&</sup>lt;sup>2</sup> Dettmer Group GmbH & Co. KG., 28195 Bremen, Germany

<sup>&</sup>lt;sup>3</sup> School of Ocean and Earth Science, University of Southampton, National Oceanography Centre Southampton, Southampton, SO14 3ZH, UK

<sup>&</sup>lt;sup>4</sup> Alfred-Wegener-Institut, Helmholtz-Zentrum für Polar- und Meeresforschung, Am Handelshafen 12, 27570, Bremerhaven, Germany

<sup>&</sup>lt;sup>5</sup> University of Bergen, Department of Earth Science and Bjerknes Centre for Climate Research, 5007, Bergen, Norway

<sup>&</sup>lt;sup>6</sup> MARUM - Center for Marine Environmental Sciences, University of Bremen, Bremen, Germany

<sup>&</sup>lt;sup>7</sup>Case Western Reserve University, Department of Earth, Environmental and Planetary Sciences, Cleveland, Ohio, USA

<sup>&</sup>lt;sup>8</sup>. iC3: Centre for ice, Cryosphere, Carbon and Climate, Department of Geosciences, UiT, The Arctic University of Norway, 9037 Tromsø, Norway

<sup>&</sup>lt;sup>9</sup> iCRAG – Irish Centre for Research in Applied Geosciences, Belfield, Dublin 4, Ireland.

*incompta*. Finally, we apply the tow calibration to core top samples from the Nordic Seas to validate the calibration for use in the paleorecord.

# 40 1 Introduction








The northern North Atlantic and the Southern Oceans are areas of strong vertical mixing and convection and as a result play a major role in air-sea gas exchange, including the release or storage of deep ocean CO<sub>2</sub> (Takahashi et al., 2009;Ezat et al., 2017;Shuttleworth et al., 2021;Rae et al., 2018). The North Atlantic, the Nordic Seas, the Labrador Sea, and Baffin Bay in particular are areas of deep-water formation and contribute to the formation of North Atlantic deep waters (NADW), a key component of the Atlantic Meridional Overturning Circulation (AMOC). Today, NADW ventilates 26% of the global deep ocean (Johnson, 2008) and carries dissolved CO<sub>2</sub> to depth making the Nordic and Labrador Seas major sinks of atmospheric CO<sub>2</sub>. The extent to which CO<sub>2</sub> is taken out of the atmosphere in these regions is dependent on several factors including temperature, the strength of the AMOC (Schmittner and Galbraith, 2008), sea-ice extent (Rysgaard et al., 2009), and biological productivity (Raven and Falkowski, 1999). Understanding the variability and magnitude of changes in surface ocean pH and CO<sub>2</sub> fluxes in the high latitudes is critical to understand its contribution to regulate the ocean carbon cycle and global climate.

Whilst the knowledge of global atmospheric CO<sub>2</sub> over the last 800 kilo annum (ka) is known with confidence from bubbles of ancient air trapped in Antarctic ice cores (Bereiter et al., 2015), our understanding of regional oceanic CO<sub>2</sub> fluxes, their magnitude and temporal variability relies on marine CO<sub>2</sub> proxy records (e.g., Martínez-Botí et al., 2015; Shuttleworth et al., 2021; Si et al., 2024). One commonly applied proxy of marine CO<sub>2</sub> relies on the boron isotopic composition (δ<sup>11</sup>B) of foraminifera shells that tracks ocean pH (Cenozoic CO2 Proxy Integration Project Foster and Rae, 2016; Consortium et al., 2023). This proxy has been extensively applied at low and mid latitudes using species from cultures, tows, and core tops where their  $\delta^{11}$ B-pH relationship has been well calibrated (Henehan et al., 2016;Raitzsch et al., 2018;Guillermic et al., 2020), and thoroughly tested against the CO<sub>2</sub> record from ice cores in areas where the ocean is in equilibrium with the atmosphere (Henehan et al., 2013; Chalk et al., 2017; de la Vega et al., 2023). However, reconstructing CO2 at high latitudes is particularly challenging due to the lower species diversity living in areas of the surface ocean where temperature is <5°C and the lack of well constrained  $\delta^{11}$ B calibrations in such species (Yu et al., 2013;Ezat et al., 2017). One such species is the non-spinose planktonic foraminifera Neogloboquadrina pachyderma. It is found in abundance at latitudes above 55-60°N and exclusively in surface waters colder than 4 °C. Considering these oceanographic regions are important areas of CO<sub>2</sub> uptake via NADW formation over the observational period, it is critical to constrain how the  $\delta^{11}$ B in this species records changes in pH and CO<sub>2</sub>.

Although the boron isotope system has a relatively well understood theoretical basis (e.g., Foster and Rae, 2016), calibrations are required to account for "vital effects" – offsets from the expected  $\delta^{11}$ B value caused by foraminiferal physiology (e.g., Henehan et al., 2016; Hönisch et al., 2003). In general, geochemical calibrations rely on at least one of these three approaches: (1) the culturing of foraminifera in a controlled laboratory environment (e.g., Gray and Evans, 2019; Lea et al., 1999), (2) the collection of foraminifera from Holocene-aged core tops or sediment traps (e.g., Anand et al., 2003; Erez and Honjo, 1981; Dekens et al., 2002) and (3) the

collection of living specimen using plankton tows (e.g., Martínez-Botí et al., 2011;Mortyn and Charles, 2003;Winkelbauer et al., 2023). Whilst the culturing approach has the merit of knowing the exact seawater chemistry in which the foraminifera grew, it doesn't fully represent the environment experienced in the open ocean, such as ontogenetic vertical migration (Meilland et al., 2021), changes in biological productivity, nutrient abundance, sea ice cover, or secondary diagenetic processes such as precipitation or dissolution in the water column and sediments. Culturing foraminifera is also a time consuming and specialist activity requiring ready access to live foraminifera and careful maintenance of culture conditions, which can be challenging. The approach using core top samples accounts for all secondary alterations of the primary signal as the foraminifera is incorporated in the sedimentary archive and is thus well-suited for application in the paleoclimatic record. However, it is challenged by the difficulty to find a natural modern range of upper ocean pH (reflected by the  $\delta^{11}$ B of borate) that is sufficiently large to define a reliable calibration (e.g., Yu et al., 2013) and there remains considerable uncertainty in determining the seawater carbonate chemistry the foraminifera calcified in, especially in regions like the northern North Atlantic where there has been significant invasion of anthropogenic CO<sub>2</sub> (Yasunaka et al., 2023). Finally, there can be uncertainty in core top sample age making the determination of seawater temperature and anthropogenic carbon contribution more uncertain.

To date, only one study has attempted to construct a  $\delta^{11}$ B calibration (Yu et al., 2013) using *N. pachyderma* from core top samples from the Labrador and Irminger Sea covering a relatively narrow range of pH and assuming a constant habitat depth of 50 m. Several attempts have been made to constrain the habitat depth of *N. pachyderma* and it is often stated that this species is a subsurface dweller (e.g., Kohfeld et al., 1996;Simstich et al., 2003) that is migrating to deeper waters prior to calcifying a thick outer crust. However, more recent direct observations of *N. pachyderma* habitat across the Arctic and polar oceans show that the habitat depth of *N. pachyderma* is variable, can be shallow (i.e. < 20 m) and is primarily controlled by sea ice cover and chlorophyll intensity (Pados and Spielhagen, 2014;Greco et al., 2019) and not by diel vertical migration or the lunar phase (Greco et al., 2019). When sea-ice cover is reduced and/or when chlorophyll at the surface is low, the habitat generally deepens to 75–150 m (Greco et al., 2019). Critically, these studies have shown that heavily crusted *N. pachyderma* specimens can be found at any depth including the top 0-50 m (Tell et al., 2022) suggesting that there is no systematic ontogenetic vertical migration associated with reproduction or "crusting" for *N. pachyderma* (Manno and Pavlov, 2014;Tell et al., 2022). The considerable variability in habitat depth across the top 200 m of the surface ocean complicates the attribution of a single depth to core top calibrations and potentially results in large uncertainties when attributing modern pH and thereby  $\delta^{11}$ B of borate to foraminiferal  $\delta^{11}$ B.

To avoid the assumptions associated with traditional core top calibrations we present here a calibration using living N. pachyderma and N. incompta specimens (e.g., cytoplasm intact, no crust) collected via plankton tows from across a large range of pH in the Labrador Sea, Baffin Bay, and the Nordic Seas. This approach allows us to compare the foraminiferal  $\delta^{11}$ B with the pH of seawater and hence  $\delta^{11}$ B of seawater borate from the exact same depth at which the tows of N. pachyderma and N. incompta were collected. While this approach provides the most accurate hydrographic data for the calibration effort, the tow depth range, in some cases, integrates a large gradient in seawater carbonate chemistry and temperature, which introduces some uncertainties (reported in Table 2). We acknowledge that the depth of the plankton tows does not necessarily represent the depth of calcification. Further,

the  $\delta^{11}B$  composition of non-crusted specimens analysed here may be different to crusted specimen found in marine climate archives. In order to assess the validity of the calibration we construct, we then apply the tow-based calibration to a series of high latitude core tops alongside existing data from the literature to evaluate its application to the paleorecord.



Whilst N. pachyderma and N. incompta were historically considered as two morphotypes of the same species (N. pachyderma sinistral and dextral respectively), they are now considered two separate species based on genetics, biogeography and ecological distinctions (Darling et al., 2006;Cifelli, 1961). They are however closely related; both are non-spinose species and live in polar or subpolar environments. N. pachyderma is typically found in cold open ocean waters and is the only species present in waters below  $4 \,^{\circ}$ C (e.g., Baffin Bay, Labrador Sea, Greenland Sea) and N. incompta dominates warmer subpolar waters of the eastern Norwegian Seas, and subpolar North Atlantic. The large geographical area of the polar North Atlantic allows us to cover a wide range of temperature, convection and  $CO_2$  flux, and as a result a large gradient of pH in this calibration.





#### 2 Methods

## 2.1 Oceanographic setting

The dataset presented here was collected from the Nordic Seas, constituting the Greenland, Icelandic and Norwegian Seas (also sometimes called in conjunction with the Arctic Sea, the Arctic Mediterranean Sea), and the Labrador Seas. At the surface, the North Atlantic Drift enters the Nordic Seas (as "inflow") mainly through the Iceland-Scotland ridge, and to a lesser extent the Denmark Strait (Hansen and Østerhus, 2000;Østerhus et al., 2019). North of the Iceland-Scotland ridge the North Atlantic Drift becomes the Norwegian Current, a warm and saline current that gradually cools via air-sea exchanges as it flows northward. From the Arctic Ocean surface outflow through the Fram Strait feeds the East Greenland Current, and through the Canadian Archipelago the Baffin Island Current. Once in the Nordic Seas the East Greenland Current follows the Greenland coast and becomes the West Greenland Current entering the Labrador Seas and subsequently the Baffin Bay. This current partly mixes with the Irminger current, a branch of the North Atlantic Drift, which increases its salinity. Together, the West Greenland and Irminger Currents flow northward into the Baffin Bay before recirculating southward alongside the surface Labrador Current. The Arctic waters that enter the Baffin Bay through the Canadian Archipelago travel via various sounds and straits southward and are of surface and subsurface origin (Azetsu-Scott et al., 2010). Along the way they mix with low salinity waters from precipitation, river, and sea ice meltwater before contributing to the low-density Baffin Island Current on the Western margin of the bay. The Baffin Island Current then mixes with the outflow water from the Hudson Bay and merges with the Labrador Current.

This complex system of water masses influences seawater pH in all regions. Mainly, the cold waters of Arctic origins are characterized by low pH due to the strong temperature dependent solubility of CO<sub>2</sub> at lower temperatures. Conversely, the warmer and saline waters feeding the Norwegian margin, the Labrador Sea, and the southern part of the Baffin Bay are characterized by higher pH. These strong gradients in surface water pH also result in a large range of CO<sub>2</sub> offsets between the surface ocean and the atmosphere (Figure 1). As a result, this

wide geographic spread and diversity in water masses allows for the large range of > 0.2 seawater pH (Table 2) and diverse foraminifera habitat included in this calibration effort.

Figure 1. Map of the delta partial pressure of CO<sub>2</sub> (pCO<sub>2</sub>, in units of micro atmospheres) defined as the difference between atmospheric and seawater pCO<sub>2</sub> (= pCO<sub>2 atmosphere</sub> – pCO<sub>2 seawater</sub>) at sea surface temperature (Takahashi et al., 2019) and location of stations where plankton tows (black circles) and core top samples (light blue crosses) were collected. Delta pCO<sub>2</sub> for the Baffin Bay was calculated using pCO<sub>2</sub> values reported in (Nickoloff et al., 2024). Map made with Ocean Data View (Schlitzer, 2002).

# 2.1. Sample material





155

Specimens collected via plankton tows include 15 N. pachyderma samples from the Labrador Sea and Baffin Bay collected during RV Maria S. Marian cruises MSM09 in 2008, MSM44 in 2015, and MSM66 in 2017. In addition, 5 tows containing N. incompta and 5 core top samples (containing N. pachyderma) were collected from the Nordic Seas during cruise CE20009 in August 2020 (Figure 1). Towed samples were collected from 100 µm mesh multinets, brought on board, and either immediately frozen at -80 °C or immediately picked into slides and frozen at -80 °C to preserve the foraminiferal cytoplasm. Frozen tows were thawed onshore and between 0.7 and 2 mg of uncrusted specimens with intact cytoplasm were hand-picked for  $\delta^{11}B$  and trace elemental to calcium analysis. The average size fraction of tow samples from the Labrador Sea ranged between 182-248 µm determined using a Keyence HX 6000 digital microscope, however all specimen sizes available were included in analysis including specimen >100 and <300 μm to ensure enough material was available. Core tops were collected from multicores recovered during cruise CE20009 (September 2020) and cruise CAGE-ARCLIM (June 2022). The multicores were immediately processed onboard and sliced at 0.5 to 1 cm resolution, and subsequently frozen at -20 °C. On land, multicore tops were thawed or freeze-dried and wet-sieved at 63 µm. 1-2 mg of N. pachyderma were picked in the 200-250 µm size fraction. When possible morphotype II was favoured when picking specimens from core tops to minimize potential depth habitat variability with morphotypes following the classification of Altuna et al. (2018, Plate 1). All specimens from core tops were encrusted and while we selected specimens from a very narrow size fraction, we did not perform a systematic analysis of the degree of encrustation of N. pachyderma in the core

top samples. Therefore, we cannot exclude that post-mortem depositional processes including the precipitation of inorganic calcite, dissolution, or recrystallization impact the thickness or geochemical composition of the crust as the tests sink through the water column and settle at the seafloor. In Table S4 we list core tops that were radiocarbon dated. Accelerator mass spectrometry (AMS) radiocarbon (<sup>14</sup>C) was measured at the Keck Carbon Cycle Accelerator Mass Spectrometry facility at UC Irvine, USA. The samples for radiocarbon dating consist of the planktonic foraminifera *N. pachyderma*. Individuals were picked at from the 0.0-0.5cm interval for each core top (Table S4).








185

## 2.2. Estimates of $\delta^{11}$ B of borate

CTD (conductivity, temperature, depth) casts with Niskin bottles were deployed at each station for cruise CE20009 and water samples were collected for total alkalinity (TA) and dissolved inorganic carbon (DIC) analysis (described in Morley et al., 2024). This allows the carbonate chemistry of seawater to be fully constrained at exactly the same depth as where the tows were collected.  $\delta^{11}B$  of borate was calculated from pH (derived from TA and DIC) and derived using the following equation:

$$\delta^{11} B_{B(OH)_{4}^{-}} = \frac{\delta^{11} B_{sw}[B]_{sw} - \epsilon_{\textit{B}}[B(OH)_{3}]}{[B(OH)_{4}^{-}] - \alpha_{B}[B(OH)_{3}]}) \text{ (eq. 1)}$$

Where the total boron in seawater  $[B]_{sw} = 432.6 \times (salinity/35) \, \mu mol/kg$  (Lee et al., 2010),  $\epsilon_B$  is the fractionation factor between the two boron species expressed as  $\epsilon_B = (\alpha_B-1) \, x \, 1000$ , the isotopic fractionation factor  $\alpha_B$  between  $B(OH)_3$  and  $B(OH)_4$  is 1.0272 as determined by Klochko et al. (2006) and the  $\delta^{11}B$  of seawater is 39.61 % (Foster et al., 2010). For stations in the Labrador Sea and the Baffin Bay the water chemistry was determined using nearby cruise data collected at the same time or the same month of the year as the tow samples (Table S3). For the towed samples, the  $\delta^{11}B$  of borate was determined using the median of the hydrographic parameters within the towed sample depth interval.

For core top samples we estimate the depth of  $\delta^{11}B$  of borate using the traditional core top approach to constrain calcification depth based on paired  $\delta^{18}O$  measurements (e.g., Ravelo and Hillaire-Marcel, 2007). Briefly, we constrained calcification depth by projecting the  $\delta^{18}O$  of N. pachyderma and N. incompta onto the equilibrium  $\delta^{18}O$  of calcite in the water column ( $\delta^{18}O_{\text{calcite.eq.PDB}}$ ) using modern hydrographic profiles and the paleotemperature equation derived by Shackleton (1974) as follow:

$$\delta^{18}O_{calcite.eq.PDB} = \delta^{18}O_{sw.SMOW} - (\frac{4.38 - \sqrt{4.38^2 - 4*0.1(16.9 - T)}}{0.2}) \text{ (eq. 2)}$$

Seawater temperature in degrees Celsius (T),  $\delta^{18}O_{calcite.eq.PDB}$  is the equilibrium  $\delta^{18}O$  of calcite on the PDB scale and  $\delta^{18}O_{sw.SMOW}$  is the  $\delta^{18}O$  of seawater on the SMOW scale (Table S1), either directly measured from CTD bottles or derived from salinity using the regional salinity— $\delta^{18}O_{sw}$  of Simstich et al. (2003). This approach yields a depth estimate for the core top foraminifera between 25 and 91 meters and an average of 65 meters (Morley et al., 2024, Supplementary Table 4). As in Morley et al. (2024) we did not apply a systematic correction for a vital effect on  $\delta^{18}O$  values because there isn't consensus in the literature on the offset recorded in *N. pachyderma* collected from core top samples. Briefly, the negative  $\delta^{18}O_c$  offset from equilibrium measured in living (uncrusted) *N*.

pachyderma collected from plankton tows is highly variable within a region, across regions, seasons, and on interannual timescales (Stangeew, 2001; Volkmann and Mensch, 2001; Pados and Spielhagen, 2014; Livsey et al., 2020). Furthermore, the addition of crust, typical for specimens recovered from core tops, increases the  $\delta^{18}O_c$  signal because crust is isotopically heavier than ontogenetic calcite (Kozdon et al., 2009; Mikis et al., 2019). Variations in the degree of encrusting can therefore either "lower", completely "mask", or shift the vital effect towards positive values depending on the degree of encrustation (Kozdon et al., 2009). As a result, there remains considerable uncertainty in the exact value to use when correcting for the competing signals of vital effects and crust in *N. pachyderma*, with studies reporting both calcification in equilibrium (Jonkers et al., 2010; Jonkers et al., 2013; Mikis et al., 2019; Jonkers et al., 2022) and out of equilibrium with seawater (Kohfeld et al., 1996; Bauch et al., 1997; Simstich et al., 2003; King and Howard, 2005). By not applying a correction for vital effects or the contribution of crust calcite we avoid biasing calcification depth estimates towards deeper depths and lower  $\delta^{11}B_{borate}$  values. This approach is also consistent with sediment trap studies from both the North and South Atlantic where  $\delta^{18}O_c$  values measured on crusted *N. pachyderma* are in equilibrium with measured water column  $\delta^{18}O$  (Jonkers et al., 2013; Mikis et al., 2019).







225

230

The pH and  $\delta^{11}B_{borate}$  for core tops samples was calculated by applying a correction for anthropogenic carbon on DIC (using the anthropogenic DIC of GLODAPv2. (2016)) ranging ~35 to 50 µmol/kg in the Nordic Sea stations in the upper 200 meters (Lauvset et al., 2022). For core tops dated with AMS radiocarbon dates younger than 550 years we did not apply a correction for anthropogenic carbon (Table S4). The uncertainty for calcification depth was defined as the minimum and maximum within a  $\pm$  20-meter interval around the central value. We note that this approach yields an average depth of the foraminifera population signal, and that despite both species being observed to live throughout the upper 200 m of the water column (Greco et al., 2019), we assume the inferred depth is representative of where the majority of the population lives and calcifies. The  $\delta^{11}B_{foram}$  and corresponding  $\delta^{11}B_{borate}$  were fitted with a York regression (York et al., 2004) that accounts for the uncertainty in both the x and y axis.

## 2.3. Elemental and δ<sup>11</sup>B analysis

Analysis was conducted in two laboratories at the University of Southampton (for samples from CE20009 in the Nordics Seas) and the Alfred Wagner Institute (AWI, for tows from the Labrador Sea and Baffin Bay). Standards and reference material are presented in both laboratories to showcase reproducibility.

# 2.3.1 Samples from cruise CE20009

Foraminifera from tows and core tops were cleaned using a modified version of the established foraminifera method to account for higher organic content in plankton tow samples (Henehan et al., 2016;Barker et al., 2003). All foraminifera were gently crushed between two glass slides to facilitate cleaning and clay removal was carried out through sequential Milli-Q rinses and brief ultrasonication to agitate the samples (core top samples only). Both core top and tow samples were oxidatively cleaned with a solution of  $H_2O_2$  (30% by weight) buffered with 0.1M NH<sub>4</sub>OH (3.4% of peroxide in the final oxidative mix). Samples were placed in a hot bath for 3x5 min for core tops and 3x20 min for tows to ensure the removal of additional organics, separated by brief 15-seconds ultrasonication. Samples were then weak acid-leached (in 0.0005 M HNO<sub>3</sub>) for 30 seconds and dissolved in ~300

 $\mu$ l of 0.15 M HNO<sub>3</sub>. An aliquot of 20  $\mu$ l (diluted in 130  $\mu$ l 0.5 M HNO<sub>3</sub>) was kept for elemental analysis, and the remainder kept for boron sample preparation and analysis.

Elemental analysis was performed on an Thermo Element Inductively Coupled Plasma Mass Spectrometer (ICP-MS) at Centre for Earth Research and Analysis Southampton (CERAS) at the University of Southampton. Element-to-calcium ratios were measured against <sup>43</sup>Ca and <sup>48</sup>Ca, then averaged, and referenced against in house mixed element standards. Consistency standards placed at the beginning and end of each sequence were measured at the same concentration as samples to assess accuracy. For samples below one mmol/mol, Element/Ca ratios typically drift, therefore, the concentration of the samples and standards were matched to account for this effect. El/Ca ratios measured included Mg and Ba. Based on the reproducibility of our in-house standards, the uncertainty for most elemental ratios is ~5 % (at 95 % confidence; (Henehan et al., 2015)). In order to showcase reproducibility between coretops measured at Southampton and tows at AWI (section 2.3.2), we report the long-term averages for Mg/Ca and Ba/Ca for NIST-C (coral reference standard) typically measured at Southampton, and the long-term average for JCP-1 typically measured at AWI and compare these to the interlaboratory assessment published in Stewart et al. (2021) and Hathorne et al. (2013), respectively (Table 1). This comparison shows that both laboratories produce consistent values without any significant interlaboratory offsets.

Table 1: Comparison of El/Ca standards measured at CERAS Southampton and AWI






| Standards | <b>NIST</b> RM 8301 | (Coral)                              | JCP-1 (uncleaned)        |                    |                                          |
|-----------|---------------------|--------------------------------------|--------------------------|--------------------|------------------------------------------|
| El/Ca     | Stewart et al. 2021 | CERAS<br>Southampton<br>(this study) | Hathorne et al. 2013     | AWI (this study    | CERAS Southampton (Stewart et al., 2016) |
| Mg/Ca     | 4.11<br>(RSD=±0.20) | 4.20<br>(SD=±0.04)                   | 4.20 (Robust SD=±0.065)  | 4.05<br>(SD=±0.13) | 4.14 (SD=±0.04)                          |
| Ba/Ca     | 5.92<br>(RSD=±0.16) | 5.71<br>(SD=±0.03)                   | 7.47(Robust<br>SD=±0.66) | 7.00<br>(SD=±0.48) | n/a                                      |

Samples for boron isotope analysis were first purified for boron by anion exchange chromatography in the boron-free class 100 laboratory of CERAS following the procedure of Foster (2008). A total procedure blank (TPB) was conducted for all batches of ~10 columns and ranged 31 to 92 pg (equivalent to 1-5% of the sample total boron). Samples were subsequently corrected using a long term  $\delta^{11}B$  of TPB of -7.27 ‰ from long term measurement at the University of Southampton. This gave a  $\delta^{11}B$  correction of 0.1 to 1.1 ‰. To test the effect of TPB correction on the regression slope, we present  $\delta^{11}B$  both with and without TPB correction.

The purified sample was analysed for  $\delta^{11}B$  at CERAS on a Thermo Neptune Multi Collector Inductively Coupled Plasma Mass Spectrometer (MC-ICPMS) with  $10^{12}\,\Omega$  amplifier resistors, using a standard-bracketing technique with NIST SRM 951 (Foster, 2008;Foster et al., 2013) to correct for the instrumental mass bias and drift. All other sample preparation and elemental analysis for the samples were conducted in the same CERAS laboratories at the University of Southampton.

The uncertainty on foraminifera  $\delta^{11}B$  is dependent on the boron content (Rae et al., 2011), i.e. the intensity of the  $^{11}B$  signal of each sample. The relationship between  $^{11}B$  signal and sample uncertainty was empirically

determined based on the uncertainty of repeated measurements of JCp-1 Coral (Porites sp.) that has undergone the same chemical purification It is defined by the following equation (Anagnostou et al., 2019):

$$2\sigma = 12960e^{-212 \times [11B]} + 0.3385e^{-1.544 \times [11B]}$$
, (eq. 3)

where [ $^{11}B$ ] is the intensity of  $\delta^{11}B$  signal in volts. Carbonate and boric acid samples were measured at CERAS and AWI and are in good agreement. The long-term  $\delta^{11}B$  value of JCp-1, AE120 and AE121 at CERAS are (at  $1\sigma$ ) 24.21  $\pm 0.12\%$ , -20.2  $\pm 0.19\%$  and 19.59  $\pm 0.28\%$  respectively, in good agreement with results of Gutjahr et al. (2021) and Stewart et al. (2021), and are comparable with values measured at AWI of 24.28  $\pm 0.3\%$  (Raitzsch et al., 2018), -19.85  $\pm 1.57\%$  (n=4) and 19.5  $\pm 0.79\%$  (n=10), respectively. Samples with substantial TPB corrections have lower confidence due to higher uncertainty in the  $\delta^{11}B$  of TPB samples (given the low boron content). Applying a long term  $\delta^{11}B$  of TPB of -7.27 % may not be representative of the true value. As a result, for samples with substantial TPB correction, the uncertainty of  $\delta^{11}B_{TPB}$  was propagated into the corrected  $\delta^{11}B$  value measured on foraminifera samples (e.g.,  $\delta^{11}B_{foram}$ ).

# 2.3.2. Samples from the Labrador Sea and Baffin Bay








Boron (B) isotope ratios for samples from the Labrador Sea were measured following Raitzsch et al. (2018). Briefly, the cleaned samples were dissolved in  $60 \,\mu\text{L}$  of  $1 \,\text{N}$  HNO<sub>3</sub> and loaded on a Savillex teflon lid then closed and turned upside to be micro-distilled on a hotplate to separate boron from the carbonate matrix. The micro-distillation method in Ca-rich matrix samples such as foraminifera yields a B recovery of ~100 %, a low procedural blank, and accurate results, even at low B concentrations (Gaillardet et al., 2001;Wang et al., 2010;Misra et al., 2014;Raitzsch et al., 2018). The procedural blank contribution during this study was  $10 - 50 \,\text{pg}$  B, which equates to ~  $0.2 \,\%$ – $0.8 \,\%$  of the total [B] in the micro-distillation vial. The distillate containing only boron was diluted with 2% HNO<sub>3</sub> and analysed for boron isotopic composition in triplicate using a Nu Plasma II multi-collector inductively coupled plasma mass spectrometry (ICPMS) at AWI (Bremerhaven, Germany) that is equipped with a customized detector array of  $16 \,\text{Faraday}$  cups and  $6 \,\text{secondary}$  electron multipliers (SEM), also termed ion counters (IC), where high-mass IC5 was used for  $^{11}\text{B}$  and IC0 for  $^{10}\text{B}$ .

Similar to coretop samples measured at Southampton, samples were measured for  $\delta^{11}B$  using a standard-sample-bracketing technique frequent analysis of control standard AE121 with an isotopic composition similar to that of foraminifera was monitored to ensure measurement accuracy. Each micro-distilled sample was analysed in triplicate where at least two measurements were used for averaging the  $\delta^{11}B$  value measured. Measurement uncertainties are reported as 2 standard deviations (2 $\sigma$ ) derived from triplicate measurements or as  $\pm$  0.30 % determined from long-term reproducibility (2 $\sigma$ ) of the control standard, whichever is larger. For elemental to calcium (El/Ca) ratios the Labrador Sea and Baffin Bay samples were analysed using a Nu AttoM high-resolution double-focusing inductively coupled plasma mass spectrometer (ICP-MS) at the AWI.

# 2.4. Core top samples

To evaluate the validity of the tow-based  $\delta^{11}$ B-pH calibration the core tops were considered as unknown paleo samples by applying the calibration equation derived from the plankton tows together with the revised Mg/Ca temperature calibration recently published by Morley et al. (2024). We applied this scheme to core tops from

various location in the North Atlantic, cruise CE20009 and CAGE-ARCLIM (Nordic seas, this study), RAPID-35-25B (North Atlantic, (Yu et al., 2013;Moffa-Sánchez et al., 2014)), and JM-FI-19PC (Norwegian sea, (Ezat et al., 2017)). The foraminiferal  $\delta^{11}B$  data from Ezat et al. (2017) measured with negative thermal ionization mass spectrometry (N-TIMS) was corrected for offsets in analytical techniques (using a correction of -1‰ defined by the foraminifera data of Farmer et al. (2016) to align with our data sets measured with MC-ICPMS. We calculate  $\delta^{11}B$ -derived pH (eq 1.) and atmospheric CO<sub>2</sub> and correct for CO<sub>2</sub> differences between foraminifera habitat depth and sea surface (from CTD or GLODAP v2 pCO<sub>2</sub> profiles) as well as local air-sea CO<sub>2</sub> disequilibrium (Takahashi et al., 2019), and compare our result to the ice core CO<sub>2</sub> record (Bereiter et al., 2015) over the last 500 years. Unlike comparing foraminiferal  $\delta^{11}B$  core top values to modern pH and  $\delta^{11}B$ borate which does not account for possible sediment age variations, this approach allows for: (1) comparison of  $\delta^{11}B$ -derived CO<sub>2</sub> with an independent record from the ice core; and (2) the knowledge of CO<sub>2</sub> variations in the recent past to evaluate the range of CO<sub>2</sub> (and pH) possibly recorded by the  $\delta^{11}B$  of core top foraminifera of variable ages.

For this calculation we use the constant fractionation factor determined by Klochko et al. (2006); eq. 1). We use a  $\delta^{11}$ B of seawater of 39.61 % (Foster et al., 2010), and use alkalinity as a second carbonate parameter in the CO<sub>2</sub> calculation (Zeebe and Wolf-Gladrow, 2001). Aqueous CO<sub>2</sub> was determined as follow:

$$CO_2 = \frac{{}^{TA} - \frac{{}^{K}_{B} \cdot B_{T}}{{}^{K}_{B} + [H^+]} \frac{{}^{K}_{W}}{[H^+]} + [H^+]}{\frac{{}^{K}_{1}}{[H^+]^2} + \frac{{}^{2}{K}_{1} \cdot K_{2}}{[H^+]^2}}$$
(eq. 4)


335

340

TA is the total alkalinity measured from CTD bottle profiles. To account for uncertainty in habitat depth TA uncertainty was estimated by integrating the total range of TA within the upper 100 meters of the water column.  $K_B$  is the equilibrium constant of boron species in seawater (a function of temperature T, salinity S and pressure P, (Dickson, 1990)),  $B_T$  the concentration of boron in seawater (432.6  $\mu$ mol kg<sup>-1</sup>, (Lee et al., 2010), [H<sup>+</sup>] the concentration of H<sup>+</sup> determined from pH =  $-\log[H^+]$  (total scale),  $K_w$  the dissociation constant of water (function of T, S and P), and  $K_1$  and  $K_2$  the first and second dissociation constants of carbonic acid (function of T, S and P, (Sulpis et al., 2020)). The partial pressure of pCO<sub>2</sub> (in parts per million, ppm) is determined as:

$$pCO_2 = [CO_2]_{sw}/K_0$$

360

365

355

With  $[CO_2]_{sw}$ , the aqueous  $CO_2$  concentration (eq. 4), and  $K_0$  Henry's law constant (Weiss, 1974). Following Martínez-Botí et al. (2015) and Chalk et al. (2017), the p $CO_2$  uncertainty was determined with a Monte Carlo simulation (10,000 realisations) to account for the uncertainty of all input parameters (all with a normal distribution). An error envelope was calculated at 1 and 2  $\sigma$  based on the 68% and 95% distribution of all the realisations.

# 3-Results

#### 3.1 Tow samples

The  $\delta^{11}B$  of towed *N. pachyderma* and *N. incompta* samples from the Labrador and Nordic Sea (i.e.  $\delta^{11}B_{foram}$ ) are compared to seawater  $\delta^{11}B$  borate (e.g.,  $\delta^{11}B_{borate}$ ) in Figure 2a. The  $\delta^{11}B_{foram}$  of both species overlaps and both are offset to lower  $\delta^{11}B$  than ambient borate. The regression for these data, using a constant  $\alpha_B$  (Klochko et al., 2006) is best described by the following equation (with  $1\sigma$  uncertainty, Figure 2a, Table 3), with a mean square weighted deviation (mswd) of 0.58 (and p=0.28):

$$\delta^{11}B_{foram} = 1.58 (\pm 0.38) * \delta^{11}B_{borate} - 11.09 (\pm 5.9) (n=16)$$

The slope of 1.58, being larger than unity, shows that at lower ambient  $\delta^{11}B_{borate}$  (lower pH), the  $\delta^{11}B_{foram}$  is more offset from equilibrium than at higher  $\delta^{11}B_{borate}$ . Furthermore, the slope lies below the 1:1  $\delta^{11}B_{foram}$ :  $\delta^{11}B_{borate}$  line. When assessing *N. pachyderma* samples by themselves (i.e. tows from the Labrador Sea) the slope is steeper at 1.82. However, both slopes and intercepts are within error of each other (Table 3). We do not attempt to define a slope for *N. incompta*-only due to the narrow range of pH and  $\delta^{11}B_{borate}$  covered for these samples. The effect of TPB correction on  $\delta^{11}B$  and the regression slope (Supplementary Figure S1, Table S2) shows that the slope remains >1 when samples with high TPB are not included in the regression (Figure S1b) or not TPB corrected (Figure S1c). Elemental data Ba/Ca of *N. pachyderma* and *N. incompta* show highly variable values ranging 2 to 602 µmol/mol (Figure 4, Table 2).

Table 2. Hydrographic dara,  $\delta^{11}B_{toram}$  and B/Ca measured on living N. pachyderma and N. incompta from plankton tows. See also Fig. 2

| المعدد حاليا مادها طالباء معاما | 0 (5 100 0 100 0 | Toralli             |           |               |        | ,  |        |       |          |         |                       | 0.                |      |                                  |            |                                      |
|---------------------------------|------------------|---------------------|-----------|---------------|--------|----|--------|-------|----------|---------|-----------------------|-------------------|------|----------------------------------|------------|--------------------------------------|
| Region                          | Cruise           | Station             | Tow depth | Species       | Depth  | р  | Hd     | Te    | Temp Sal |         | $\delta^{11} B_{bor}$ |                   | 2    | $\delta^{11}$ B <sub>foram</sub> |            | Ba/Ca                                |
|                                 | ♀                | Q                   | [m]       | 1             | [m] 1c | 10 | 10     |       | [°C]     | ] [nsd] | [‰] e                 | err (lw) err (up) |      | [‰]                              | err (±) [μ | $err(\pm)$ [µmol.mol <sup>-1</sup> ] |
| NE Norwegian Sea                | CE20009          | 2a                  | 0-10      | N.incompta    | 5      | 5  | 8.14   | NA    | 10.7     | 34.44   | 17.12                 | 0.29              | 0.29 | 0.29 15.01*                      | 2.00       | 602                                  |
|                                 | CE20009          | 2b                  | 10-20     | N.incompta    | 15     | 2  | 8.16   | AA    | 10.6     | 34.63   | 17.37                 | 0.04              | 0.04 | 0.04 15.61*                      | 2.00       | 526                                  |
|                                 | CE20009          | 2c                  | 20-30     | N.incompta    | 25     | 5  | 8.16   | NA    | 10.2     | 34.80   | 17.29                 | 0.04              | 0.04 | 0.04 14.8*                       | 2.00       | 531                                  |
| Fram Strait                     | CE20009          | 5                   | 0-40      | N.incompta    | 20     | 20 | 8.14   | NA    | 7.5      | 34.90   | 16.80                 | 0.56              | 0.33 | 0.33 15.34*                      | 2.00       | 329                                  |
| Davis Strait                    | MSM44            | GeoB19913-2 0-100   | 0-100     | N.pachyderma  | 20     | 20 | 8.12 ( | 600.0 | 0.1      | 32.48   | 15.70                 | 0.40              | 0.10 | 0.10 14.04                       | 06.0       | 2.97                                 |
|                                 | MSM44            | GeoB19913-2 100-200 | 100-200   | N.pachyderma  | 150    | 20 | 8.04 ( | 0.033 | -1.2     | 33.51   | 15.13                 | 0.15              | 0.10 | 13.00                            | 0.82       | 2.93                                 |
| Northern Baffin Bay MSM44       | MSM44            | GeoB19929-3 80-100  | 80-100    | N.pachyderma  | 06     | 10 | 8.00   | 0.001 | -1.1     | 33.76   | 14.89                 | 0.02              | 0.10 | 11.59                            | 1.15       | 2.05                                 |
|                                 | MSM44            | GeoB19929-3 100-200 | 100-200   | N.pachyderma  | 150    | 20 | 7.95 ( | 0.024 | -0.7     | 33.94   | 14.67                 | 0.15              | 0.10 | 12.21                            | 0.97       | 2.02                                 |
| Southern Baffin Bay             | MSM09/2          | MSM09/2 GeoB455     | 80-100    | N.pachyderma  | 90     | 10 | 8.05 ( | 0.009 | -1.1     | 33.54   | 15.20                 | 0.11              | 0.10 | 11.76                            | 0.87       | 2.90                                 |
| Southern Labrador               | MSM09/2          | MSM09/2 GeoB415     | 20-40     | N.pachyderma  | 30     | 10 | 8.11 ( | 0.005 | 1.5      | 32.23   | 15.79                 | 0.02              | 0:30 | 13.05                            | 0.51       | 10.5                                 |
| Sea                             | MSM09/2          | MSM09/2 GeoB415     | 40-60     | N.pachyderma  | 20     | 10 | 8.09   | 0.016 | -0.7     | 33.30   | 15.48                 | 0.02              | 0:30 | 13.38                            | 0.48       | 20.6                                 |
| Labrador Sea                    | MSM09/2          | MSM09/2 GeoB433     | 100-150   | N.pachyderma  | 125    | 25 | 8.01 ( | 0.010 | 4        | 34.70   | 15.48                 | 0.02              | 0:30 | 13.74                            | 0.38       | 17.2                                 |
|                                 | MSM09/2          | MSM09/2 GeoB434     | 0-20      | N.pachydermad | 10     | 10 | 8.05   | 0.001 | 8.8      | 33.31   | 16.08                 | 0.02              | 0:30 | 14.11                            | 0.51       | 15.0                                 |
|                                 | MSM09/2          | MSM09/2 GeoB434     | 20-40     | N.pachyderma  | 30     | 10 | 8.12 ( | 90000 | 4.8      | 33.79   | 16.28                 | 0.02              | 0:30 | 14.62                            | 0.55       | 20.8                                 |
|                                 | MSM09/2          | MSM09/2 GeoB435     | 100-150   | N.pachyderma  | 125    | 25 | 8.04 ( | 0.000 | 4.1      | 34.70   | 15.65                 | 0.02              | 0:30 | 13.86                            | 0.17       | 6.46                                 |
| Baffin Bay                      | MSM09/2          | MSM09/2 GeoB460     | 40-60     | N.pachyderma  | 20     | 10 | 8.08   | 0.004 | -1.4     | 33.07   | 15.33                 | 0:30              | 0:30 | 12.82                            | 0.36       | 9.44                                 |
|                                 |                  |                     |           |               |        |    |        |       |          |         |                       |                   |      |                                  |            |                                      |

\* Total procedure blank corrected

Figure 2. Relationship between the δ¹¹B of towed N. pachyderma (black circles) and N. incompta (yellow circles) with the δ¹¹B of seawater borate. Panel a shows δ¹¹B of towed specimens against δ¹¹B of seawater borate without a temperature correction on αB on the calculation of δ¹¹B of borate. We fitted the data with a York regression (black line) and a 95% confidence envelope (light yellow shade), the x and y error bars represent the borate δ¹¹B uncertainty and the foraminifera δ¹¹B external uncertainty respectively (see methods). The 1:1 line is shown by the black dashed line. Panel b shows calibration equations for other planktonic foraminifera next to the tow-based calibration equation derived here for N. pachyderma. Panels c and d show the regression between δ¹¹B<sub>foram</sub>-δ¹¹B<sub>borate</sub> offset and temperature (n=16, p=0.07, r²=0.22) and salinity (n=16, p=0.25, r²=0.10) respectively. The blue envelope represents the 95% confidence interval of the linear regression.

**Table 3**. Boron Isotope Calibration equation for *N. pachyderma* and *N. incompta*, and corresponding mean squared weighted deviation (mswd). Slopes and intercepts were fitted with York linear regressions of the type: y=mx+c.

| Sample Type                        | Slope( $\pm 1\sigma$ ) | Intercept( $\pm 1\sigma$ ) | mswd |
|------------------------------------|------------------------|----------------------------|------|
| Tows (N. pachyderma + N. incompta) | $1.58(\pm 0.38)$       | -11.09(±5.9)               | 0.58 |
| Tows (N. pachyderma only)          | $1.82(\pm 0.5)$        | -14.95(±7.77)              | 0.58 |

# 400 3.2. Relationship between $\delta^{11}$ B, temperature and salinity.

The foraminifera  $\delta^{11}$ B exhibit a significant correlation with temperature (Fig S2 S1,  $r^2$ = 0.58, p<<0.05) and salinity (Fig.S2,  $r^2$ = 0.24, p= 0.017), which is likely due to covariance between carbonate system parameters, temperature, and salinity. To evaluate the influence of temperature and salinity on the proxy system we plot them against the

offset between  $\delta^{11}B_{\text{foram}}$  and predicted  $\delta^{11}B_{\text{borate}}$  (Figures 2c and 2d) and in both cases the correlation is not significant (i.e. p<0.05).

### 3.3. Applying the calibration to core top samples.

Whilst tows provide information about the original signal imprinted in well constrained calcifying environments, the application in the paleorecord relies on sediment samples with foraminifera that incorporate the calcification depth during ontogeny and post depositional environment processes in terms of sedimentation rate, corrosiveness of porewaters etc. The tow-based foraminifera  $\delta^{11}B$  calibration equation based on temperatures calculated from the Mg/Ca- $\delta^{18}O$  correction scheme (Morley et al., 2024) yields reconstructed atmospheric pCO<sub>2</sub> concentrations that span the pre-industrial (280 ppm) to modern range (420 ppm)

415

440

410

#### 4 Discussion

N. pachyderma and N. incompta are both non-spinose asymbiotic genetically sister species inhabiting the high latitude oceans (Darling et al., 2006). While their geographic distribution in the North Atlantic overlaps slightly at temperatures between 6-10 °C (Darling et al., 2006) each species inhabits distinct environmental conditions:
warm Atlantic waters at subpolar latitudes of high pH for N. incompta, and cold polar waters of lower pH for N. pachyderma. Combining them in our study therefore allows us to cover a wider range of pH, and hence δ<sup>11</sup>B<sub>borate</sub>, improving the overall calibration. The minor difference in the calibration slopes when considering N. pachyderma alone or if both species are combined (Table 3) further support this approach.

## 4.1. Potential drivers of $\delta^{11}$ B sensitivity and alteration of microenvironment.

Our new calibration shows that the δ<sup>11</sup>B<sub>foram</sub> recorded by *N. pachyderma* and *N. incompta* is lower than the δ<sup>11</sup>B of seawater borate (1:1 line, Figure 2). This result is in agreement with other published calibrations of the nonspinose species, *Globoconella inflata*, the symbiont-bearing spinose species *O. universa* (Henehan et al., 2016), and the symbiont-barren spinose species *G. bulloides* (Martínez-Botí et al., 2015). Further we observe that the slope of the regression is >1 (Figure 4e, Table 3). This observation contrasts with previously published δ<sup>11</sup>B calibrations that show either a slope close to unity in line with the expected pH sensitivity from δ<sup>11</sup>B of borate in seawater such as that found for epifaunal benthic foraminifera (Rae et al., 2011), *G. bulloides* (Martínez-Botí et al., 2015), *O. universa* (Henehan et al., 2016) and corals *Desmophyllum dianthus* (Anagnostou et al., 2012;McCulloch et al., 2012) and *Balanophyllia elegans* (Gagnon et al., 2021), or a slope <1 like the spinose symbiont-bearing species *G. ruber* (Henehan et al., 2013) and *T. sacculifer* (Sanyal et al., 2001;Martínez-Botí et al., 2015).

Interpretations of a weaker sensitivity (i.e. slope <1) to pH for *G. ruber* and *T. sacculifer* invoke photosymbiont activity, calcification and/or foraminifera metabolism, altering the pH of the diffusion-controlled microenvironment immediately surrounding the living foraminifera (e.g. (Henehan et al., 2016;Hönisch et al., 2003;Zeebe et al., 2003). In this scenario, lower pH in the microenvironment relative to ambient seawater immediately surrounding *N. pachyderma* and *N. incompta* specimens could arise from the CO<sub>2</sub> flux from foraminiferal respiration and calcification, which is offset/compensated for by CO<sub>2</sub> drawdown by symbionts in

the photosymbiont bearing spinose foraminifera (Zeebe et al., 1999;Rink et al., 1998). Alternative explanations for the patterns seen in photosymbiont bearing foraminifera invoke the incorporation of the isotopically heavy  $B(OH)_3$  at slower calcification rates at lower pH, suggested to increase  $\delta^{11}B_{foram}$  resulting in a shallow  $\delta^{11}B_{foram}$  –  $\delta^{11}B_{borate}$  slope (Uchikawa et al., 2015; Farmer et al., 2019). Whether or not calcification rates impact  $\delta^{11}B_{foram}$  in 445 N. pachyderma in this way remains to be determined because little is known about the exact foraminiferal calcification rates in their natural environment. Inferred calcification rates based on laboratory culturing experiments and temperature dependent modelling suggest that Neogloboquadrinoids have lower calcification rates overall than other planktonic foraminifera species (Lombard et al., 2011). Furthermore, recent advances in 450 three-dimensional imaging techniques (microCT scans) propose that growth patterns of Neogloboquadrinoids through ontogeny follow a distinct growth trajectory that unlike other spinose species slows at maturity for the final whorl of chambers (Burke et al., 2020), which constitute 70% or more of the total pre-gametogenic CaCO<sub>3</sub>. Whole-shell measurements of non-crusted tows, such as those here, are likely dominated by the isotopic composition recorded in these terminal chambers. In this scenario, if boric acid incorporation at low pH is 455 responsible for <1 slopes in some species of foraminifera, it is unlikely to play a role in the Neogloboquadrinoids data presented here as low growth rates are associated with elevated  $\delta^{11}B$  above borate rather than below borate (Figure 2b; (Farmer et al., 2019; Uchikawa et al., 2015)).

In the following sections we explore additional mechanisms that could explain the lower-than-expected  $\delta^{11}$ B<sub>foram</sub> values and high sensitivity of *N. pachyderma* to low seawater borate (slope <1).

## 460 4.1.1. Foraminiferal microenvironment and calcification




Non-spinose foraminifera are generally devoid of photosynthetic symbionts; the pH of the micro-environment around such foraminifera is expected to more acidic than the ambient seawater. The respiration and calcification of foraminifera both release  $CO_2$  into the microenvironment during these processes, which have been suggested as the main drivers explaining the lower-than-expected  $\delta^{11}B_{foram}$  values in symbiont-barren species (Foster, 2008;Hönisch et al., 2003;Martínez-Botí et al., 2015;Yu et al., 2013).

To assess the calcification environment for N. pachyderma we estimated microenvironmental pH and the calcite saturation state ( $\Omega_{\text{calcite}}$ ) for our plankton tow dataset. Briefly, we derived  $\Omega_{\text{calcite}}$  at calcification depth using hydrographic profiles (TA and DIC) which yields on average  $\Omega_{\text{calcite}} = 2.2$ . One way to approximate the conditions within the micro-environment of a living foraminiferal, without turning to complex diffusion-reaction modelling (e.g., Zeebe et al., 2003), is to simply model the effect of removing TA and DIC in a 2:1 ratio to simulate the influence of calcification (c.f. Hönisch et al., 2019). Following Hönisch et al. (2019) we modelled the progressive removal of up to 200 and 400  $\mu$ mol kg<sup>-1</sup> from the ambient DIC and TA, respectively (as in Hönisch et al., 2019; Figure 3). We note that this decrease in DIC and TA may be counteracted by diffusion. Taking the  $\delta^{11}$ B values of the N. pachyderma and N. incompta at face value, this treatment implies an average pH of 7.4 in the microenvironment with  $\Omega_{\text{calcite}} = 0.4$ . This suggests that N. pachyderma calcifies in waters undersaturated with respect to calcite.

Figure 3. Theoretical effect of calcification (inducing a decrease of TA:DIC in a 2:1 ratio) on the Omega calcite and pH of seawater in the foraminifera microenvironment,

Calcification in foraminifera, like in many marine carbonates (Gilbert et al., 2022), takes place in an internal space within a fluid, the calcifying fluid, that is ultimately modified from ambient seawater carbonate chemistry (de Nooijer et al., 2014 and references therein). Benthic foraminifera have been observed to upregulate the pH in their calcifying fluid as high as 9 (de Nooijer et al., 2009) by the removal of protons via a Ca-ATPase enzymatic pump (Toyofuku et al., 2017) and/or alkalinity elevation of seawater vacuoles to facilitate calcification (Bentov et al., 2009). It is likely that DIC is also elevated in the calcifying fluid by passive CO<sub>2</sub> diffusion and active HCO<sub>3</sub>-pumping (Ujiié et al., 2023). Indeed, genes relating to these processes were recently identified in foraminifera by Ujiié et al. (2023).

Whether or not *N. pachyderma* upregulates its internal pH remains to be determined, however, active proton pumping as a mechanism for *N. pachyderma* calcification aligns well with the recent biomineralization model proposed for Mg incorporation in *N. pachyderma* outlined in Morley et al. (2024). Briefly, they demonstrated that the preferential exclusion of Mg<sup>2+</sup> into *N. pachyderma* is compromised in unfavourable calcification environments (e.g., at low seawater [CO<sub>3</sub><sup>2-</sup>] and SST >5 °C) in order to maintain energy for proton removal (see also (Evans et al., 2018;Zeebe and Sanyal, 2002). This observation invokes two important processes: (1) the generation and transport of bicarbonate and protons across the calcifying membrane and (2) that proton pumping may be less efficient for *N. pachyderma* at temperatures below 5 °C and low sea water [CO<sub>3</sub><sup>2-</sup>]. The latter process may provide an explanation for the steep  $\delta^{11}$ B<sub>foram</sub>:  $\delta^{11}$ B<sub>borate</sub> slope observed here in equation 1, considering that the samples collected from the coldest and most acidic environments lead to the largest offset from a slope = 1 (Figure 2c).




Assuming *N. pachyderma* upregulate their internal pH, the lower-than-expected  $\delta^{11}B_{foram}$  values, suggesting a calcification fluid that is undersaturated with respect to calcite, remain puzzling. Following Gagnon et al. (2021) these observations could be reconciled if we consider boric acid diffusion into the calcifying fluid as part of the biomineralization process. If the diffusion of boric acid between the calcifying fluid and the microenvironment around the foraminifera is fast, while the exchange of seawater between the two is slow, it is possible for the concentration of boric acid internally to be the same as that in the micro-environment. Since the  $\delta^{11}B$  of each

aqueous boron species is set by their relative proportions, this sets the  $\delta^{11}B$  of borate in the calcifying fluid to be equal to the  $\delta^{11}B$  of borate externally, even if the pH internally is significantly greater than the micro-environment. While this model reconciles the upregulation of pH in the calcifying fluid enabling calcification with  $\delta^{11}B_{foram}$  values representative of the (undersaturated) microenvironment surrounding the foraminifera, it awaits direct determination of the calcifying fluid pH in *N. pachyderma* to validate it.

## 4.1.2 The effect and impact of temperature on the boron isotope fractionation factor $\alpha_B$

The influence of temperature on the aqueous boron isotope fractionation factor α<sub>B</sub> has been the subject of debate (Hönisch et al., 2019) even though there is a theoretical thermodynamic basis for α<sub>B</sub> to be influenced by temperature (Zeebe, 2005), its magnitude is uncertain and it has not been conclusively demonstrated experimentally. So far, the fractionation α<sub>B</sub> has been determined at 25 and 40 °C at a salinity of 35 (Klochko et al., 2006). However, large uncertainties prevent a conclusive quantification of the temperature dependence of α<sub>B</sub>, which has been used to argue against a significant temperature sensitivity of α<sub>B</sub> over the temperature range used in most δ<sup>11</sup>B-pH calibrations and paleo-pH studies (Foster and Rae, 2016). In Figure 2, we show the calibration equation with constant α<sub>B</sub> as defined in Klochko et al. (2006) and in Figure S2c with a temperature dependent α<sub>B</sub> as defined in Hönisch et al. (2019).

Due to the large temperature gradient in our dataset,  $\alpha_B$  corrected for temperature has a significant effect on the slope and intercept (Figure S2b and S2c). Like in other symbiont-barren planktonic foraminifera the temperature correction, when applied to *N. pachyderma*, reduces the large negative offset between  $\delta^{11}B_{foram}$  and predicted  $\delta^{11}B_{borate}$  however the microenvironmental pH remains slightly below the predicted  $\delta^{11}B_{foram}$ :  $\delta^{11}B_{borate}$  line. Therefore, even with a temperature dependent  $\alpha_B$  *N. pachyderma* would still require manipulation of microenvironmental pH to reconcile the disequilibrium isotopic offset from seawater values. Furthermore, there is no significant correlation between the offsets of *N. pachyderma*  $\delta^{11}B$  from  $\delta^{11}B_{borate}$ , and temperature (Figure 2c), arguing against a strong influence of a temperature dependent  $\alpha_B$ . Thus, although a temperature dependence on the fractionation factor is likely to exist, our data suggests that it is not significant over the temperature range relevant for *N. pachyderma*.

# 4.2. The marine aggregate habitat hypothesis.







It has been recently suggested that some non-spinose foraminifera live in marine aggregate (also referred to as marine snow or particulate organic matter). This has been indirectly observed in laboratory culture for *N. pachyderma* (Davis et al., 2020), *G. truncatulinoides* (Richey et al., 2022) and from plankton net for *G. menardii* and *N. dutertrei* (Fehrenbacher et al., 2018). Geochemical analysis of the shells of these species tend to show elevated Ba/Ca (Fehrenbacher et al., 2018;Richey et al., 2022;Fritz-Endres et al., 2022;Hupp and Fehrenbacher, 2023) compared to species that have not been observed in marine aggregates, and well above Ba/Ca values expected if precipitating in seawater given coefficient partition defined from laboratory experiment (e.g., Hönisch et al., 2011). Barium sulphate BaSO<sub>4</sub> (barite) in pelagic environment is thought to precipitate via microbial activity (e.g., González-Munoz, 2003;Martinez-Ruiz et al., 2020) or through supersaturation of barium and sulphur in microenvironment that promotes precipitation abiotically (Dehairs, 1980;Bishop, 1988;Horner and Crockford, 2021 and references therein). Hence foraminifera calcifying in such a microenvironment would have their shell enriched in barium compared to the bulk seawater. Such a microenvironment could be a concern for paleo-pH

reconstructions as we expect variable pH alteration in marine aggregates. Marine aggregates dominated by algae can result in a pH increase if the signal is dominated by photosynthesis (e.g., Ploug et al., 1999) whilst aggregates dominated by organic matter degradation will release CO<sub>2</sub> and decrease pH by 0.22-0.91 (Alldredge and Cohen, 1987). Due to the technical challenges to collect intact marine aggregates from the water column and to measure pH with microelectrodes on sometimes floating aggregates, studies on pH alterations in marine aggregates are rare (Alldredge and Cohen, 1987) and further constraints are needed to quantify pH changes on multiple aggregates where *N. pachyderma* and *N. incompta* are observed to live.







**Figure 4.** Relationship between the  $\delta^{11}$ B of towed *N. pachyderma (coloured circles* with the  $\delta^{11}$ B of seawater borate in relation to calcite Ba/Ca ratio. Panel **a.** shows  $\delta^{11}$ B of towed specimens against  $\delta^{11}$ B of seawater borate while the colour gradient is showing the respective Ba/Ca values. The 1:1 line is shown by the black dashed line. Panel b. is showing the regression between  $\delta^{11}$ B<sub>foram</sub>- $\delta^{11}$ B<sub>borate</sub> offset of towed *N. pachyderma* specimens and Ba/Ca values (n=12, p=0.1, r2=0.25). The data are fitted with a linear regression (black line) and a 95% confidence envelope (blue shade). Also shown in panel b. are the high Ba/Ca values for *N. incompta* in yellow circles; please note the axis break.

If barium concentration is an accurate metric to trace this behaviour in N. pachyderma and N. incompta we would expect samples with high Ba/Ca to record a distinct pH and  $\delta^{11}B$  from samples with low-Ba/Ca samples. Figure 4a shows some samples have higher barium than what is predicted by seawater following the partition coefficient  $D_{Ba} = 0.11$  of other non-spinose foraminifera (Fehrenbacher et al., 2018), yet do not show any anomalous  $\delta^{11}B$ compared to low-Ba/Ca samples. The elevated Ba/Ca results (> 30 umol/mol) need to be caveated by the fact that samples were not treated for barite removal during the cleaning process (as in Fehrenbacher et al., 2018), as this would lead to unacceptable sample loss for  $\delta^{11}B$  measurement. The Ba/Ca ratio in N. incompta is distinct from N. pachyderma, with a range of 328-779 μmol/mol (vs. 1-21 μmol/mol for N. pachyderma). The species difference and trend of Ba/Ca values is observed in both tows and core tops. This striking contrast between the two species points towards a distinct behavioural difference between the two species, potentially suggesting N. incompta is more prone to live in marine aggregates or has a specific feeding behaviour that would favour Ba enrichment during calcification. The lack of any deviation in  $\delta^{11}$ B space for these high-Ba samples (Figure 4b) suggest that the Ba enrichment observed did not result in a pH-altered microenvironment in marine aggregates influencing the  $\delta^{11}$ B<sub>foram</sub> and instead that Ba increase may be caused by other processes (e.g. feeding behaviour). In the absence of direct evidence of aggregate environment associated with high Ba/Ca for these two species, we conclude that the observed increase in Ba/Ca, either does not result in significant pH change or if it does, it does not compromise

the δ<sup>11</sup>B proxy in the species examined here. The hypothesis that these species live in marine aggregate, and the
link with Ba-enrichment requires further investigation through direct observation of intact marine aggregates in
the water column (e.g. collected with a marine snow catcher) or newly formed in cultures, as well as direct
measurement of pH change in the interstitial fluid of the aggregates and Ba/Ca in foraminifera shells.

# 4.3. $\delta^{11}$ B-derived CO<sub>2</sub> for core top values.

A comparison of available δ<sup>11</sup>B<sub>foram</sub> values measured on *N pachyderma* from previously published cores tops (Ezat et al., 2017;Henehan et al., 2016;Yu et al., 2013) with our tow-based calibration dataset shows that there is generally good agreement between *N. pachyderma* collected from tows and core tops (Figure 5). We note here that we recalculated seawater borate for the Yu et al. (2013) dataset using GLODAP v2 at 50 m instead of CARINA that was used in the original study. We used an anthropogenic DIC correction (C<sub>ANT</sub>) on δ<sup>11</sup>B<sub>borate</sub> values for all core tops pre-dating the onset of anthropogenic CO<sub>2</sub> emissions (see Table S4 and S6), The core tops analysed in this study fall within the calibration uncertainties of the tow-based calibration equation except for Station 16. Further, the majority of the Yu et al. (2013) data falls below the calibration equation derived using the plankton tows.

There could be several reasons for the offsets between calibration studies. First, the estimated calcification depth of 50 m for these samples may be too shallow. A shift to a deeper depth habitat (e.g. 75-150 m) would reduce  $\delta^{11}B_{borate}$  and thereby result in a better overall fit with the tow dataset (Figure S3). However, in the absence of stable isotopes we cannot constrain habitat depth with better accuracy. Further, the modern water column data including the correction factor for anthropogenic carbon to estimate seawater borate may not represent the actual 595 calcification environment of N pachyderma preserved in core tops. For example, when the correction factor for anthropogenic carbon is removed for the Yu et al. (2013) dataset, the offset to the tow based calibration dataset is resolved. However, the pre-industrial ages for all but one core top in this dataset would strongly argue for the use of the correction factor for anthropogenic carbon. For station 16, it is possible that the sample from the multicore top 0-0.5 cm is modern or partially modern and should not be corrected for anthropogenic carbon. This is 600 supported by the age model for gravity core M23411 (Baumann, 2007) collected at the same location as Station 16 providing Holocene sedimentation rates of 143 years per 0.5 cm. Therefore, the top 0-0.5 cm of sediments at this station could include a significant contribution of specimens grown after the onset of fossil fuel burning. However, in the absence of a date from the multicore top we cannot confirm it.

Figure 5. Compilation of plankton tows and core top  $\delta^{11}B_{foram}$  versus  $\delta^{11}B_{borate}$ . Here we plot our tow and core top results together with all available  $\delta^{11}B_{foram}$  values measured on *N. pachyderma* (Henehan et al., 2016;Ezat et al., 2017;Yu et al., 2013). PI stands for preindustrial and  $C_{ANT}$  for anthropogenic carbon correction. The 1:1 line is shown by the black dashed line, and the solid black line shows the York regression of the tow dataset as shown in Figure 2 with the 95% confidence envelope (light yellow shade). The Yu et al., (2013) data is plotted with  $\delta^{11}B_{borate}$  values estimated at 50 m. Please view Figure S3 for  $\delta^{11}B_{borate}$  values estimated at 150 m.

Alternatively, the lower than expected  $\delta^{11}B_{foram}$  values from the Yu et al. (2013) dataset may result from different size fractions used in the analysis. Here, core tops were sieved at a narrow size fraction of 200-250 µm, while Yu et al. (2013) used a larger range of 150–250 µm. Indeed, the offsets of *N. pachyderma*  $\delta^{11}B_{foram}$  from  $\delta^{11}B_{borate}$  are much larger for the Yu et al. (2013) dataset that includes smaller specimens. However, we note that our core tops are not offset from our plankton tow dataset which includes all size fractions. This would argue against a size effect. However, the apparent size effect could be reconciled if it is placed in the context of dissolution. For example, it has been observed that small foraminifera are more affected by dissolution than larger individuals as a result of their higher surface area to volume ratio (Berger and Piper, 1972;Berger et al., 1982). The cumulative effect of smaller size fractions and partial dissolution may therefore explain the apparent offset in the Yu et al. (2013) dataset (Ni et al., 2007).

Finally,  $\delta^{11}$ B values recorded in crust calcite may be lower than in ontogenetic calcite. This has previously been suggested for *T. sacculifer*, where crust calcite is lower in  $\delta^{11}$ B relative to ontogenetic calcite because symbionts are expelled or digested before gametogenic calcification begins (Ni et al., 2007). This loss of photosymbionts decreases the local pH due to the lack of photosynthetic activity and the  $\delta^{11}$ B<sub>borate</sub> of the microenvironment would be lower than ambient seawater values. Consequently, the crust calcite will be isotopically lighter than the ontogenetic calcite (Ni et al., 2007). However, *N. pachyderma* is asymbiotic and modern ecological observations would argue against a habitat migration to deeper depth during gametogenesis (Tell et al., 2022; Manno and Pavlov, 2014; Greco et al., 2019) and therefore it is unlikely that a difference in hydrographic conditions would

be responsible for a geochemical signature difference between ontogenetic and crust calcite. These ecological observations are supported by geochemical analysis of living (no crust) and dead (crust) *N. pachyderma* specimens collected from the same plankton nets (Hupp and Fehrenbacher, 2023), *N. incompta* and *N. dutertrei* specimen grown and crusted in culture under constant temperatures (Davis et al., 2017; WestgÅrd et al., 2023; Fehrenbacher et al., 2017) and *N. pachyderma* specimen from sediment traps (Jonkers et al., 2016).








The mechanisms responsible for the geochemical offsets observed in ontogenetic and crust calcite for N. pachyderma (e.g., Mg/Ca  $\approx 15\%$  and  $\delta^{18}$ O<sub>c</sub> up to 2 ‰), remain unresolved, but are more likely linked to changes in the biomineralization process than hydrography. For example, faster calcification rates during the formation of crust calcite relative to the last whirl of chambers could result in variable kinetic fractionation during ontogeny and therefore variation in the degree of isotopically heavy B(OH)<sub>3</sub> incorporated into the foraminiferal calcite in crusted specimen (Uchikawa et al., 2015; Farmer et al., 2019). If so, the degree of crusting would be an additional factor to consider when preparing specimens for analysis. Furthermore, the more soluble ontogenetic calcite (Wycech et al., 2018), could be prone to preferential dissolution (water column and sediments) relative to crust calcite, leaving core top N. pachyderma from sites close to undersaturation depleted in  $\delta^{11}$ B values (Hönisch and Hemming, 2004; Ni et al., 2007; Seki et al., 2010). Indeed, we note that the low  $\delta^{11}B$  values from Station 16 and from some stations of the Yu et al. (2013) dataset have low bottom water  $\Omega_C < 1.25$  values (Figure 4a, Table S5). Unfortunately, calcification rates are difficult to determine in culture for N. pachyderma as temperature control, especially at low temperatures, prevents prolonged periods of observations. Furthermore, there is no systematic  $\delta^{11}$ B offset between crusted and uncrusted specimens preventing us at this stage from estimating a geochemical offset value between ontogenetic and crust calcite (e.g., Kozdon et al., 2009). Ongoing developments in  $\delta^{11}B$ single shell analysis vs MC-ICPMS laser ablation (Standish et al., 2019; Mayk et al., 2020; Raitzsch et al., 2020) N. pachyderma may provide enhanced spatial resolution to address the foraminifera test isotopic heterogeneity.

To assess whether the tow-based calibration can be used to estimate past pH and atmospheric  $CO_2$  concentrations beyond these caveats we now treat the core tops as if they were an unknown paleo-sample.  $\delta^{11}B$  derived  $CO_2$  estimates from core tops in this study (except for station 16) provide  $CO_2$  values that are consistent with atmospheric  $CO_2$  observations made post 1900 (Figure 6). These results suggest that despite the low sedimentation rates; a proportion of modern foraminifera shells capture the anthropogenic  $CO_2$  signal within the top 0-0.5 cm of surface sediments in the Nordic Seas. We also calculated atmospheric  $CO_2$  using published core tops from Ezat et al. (2017) (which has larger uncertainties due to the additional uncertainties linked to differing analytical methods) and one core top from Yu et al. (2013) that we were able to pair with Mg/Ca values from the same multicore top published in Moffa-Sánchez et al. (2014). The highest reconstructed  $CO_2$  values consistent with modern atmospheric  $CO_2$  were measured in core tops that were confirmed modern by AMS  $^{14}C$  dating (e.g., Station 12 and RAPID-35-25B). Applying the new calibration with a slope >1 to the downcore data of Yu et al. (2013), will elevate reconstructed surface seawater  $CO_2$ . As a result, regions off Iceland in the polar North Atlantic may not have served as a strong sink of  $CO_2$  to the atmosphere during the glacial and deglacial period as hypothesized in Yu et al. (2013). A detailed analysis of existing and new palaeoceanographic datasets will be the object of a forthcoming manuscript.

**Figure 6.** Atmospheric CO<sub>2</sub> concentration [ppm] derived using boron isotopes, Mg/Ca and  $\delta^{18}$ O<sub>c</sub> from CE20009 core tops collected from the Nordic Seas (except station 16). Also shown are two core tops that have paired Mg/Ca and d18Oc values for *N. pachyderma* available. One from the Yu et al. (2013) dataset that we paired with Mg/Ca and  $\delta^{18}$ O<sub>c</sub> from Moffa-Sánchez et al. (2014) and the other is from Ezat et al. (2017). The CO<sub>2</sub> uncertainty was calculated with a Monte Carlo simulation (see methods) See also Table S6.

#### 5. Conclusion

Here we present the first tow-based open ocean  $\delta^{11}B$  calibration of the non-spinose foraminifera *N. pachyderma* and *N. incompta* in the North Atlantic, complemented by core tops from similar locations. The use of tows allows us to precisely constrain the water chemistry in which foraminifera grew, bypassing the assumptions of foraminifera habitat depth and water chemistry typically encountered in core-top-based calibrations. We show that the signal recorded by these species is consistently below what is predicted by aqueous  $\delta^{11}B_{borate}$ , in line with other non-spinose foraminifera and that there is little evidence to support a strong influence of a temperature dependent  $\alpha_B$ .




We propose instead that the lower-than-expected  $\delta^{11}B_{\text{foram}}$  values are a result of active proton pumping during calcification raising the pH in the calcification fluid while the microenvironment surrounding the cell is undersaturated with respect to calcite as a result of respiration and calcification. The seemingly opposing signals can be reconciled when considering rapid boric acid diffusion between the calcifying fluid and the microenvironment around the foraminifera (Gagnon et al., 2021).

The addition of core top samples in the calibration does not significantly alter the slope of the calibration, giving confidence in the application of this tow-based calibration equation in downcore sediments. This is further supported when combining the  $\delta^{11}B$  calibration with the Mg/Ca- $\delta^{18}O$  correction scheme to reconstruct past atmospheric CO<sub>2</sub> concentrations. Given the rapid rise in recent atmospheric CO<sub>2</sub> concentrations, highly resolved marine archives when independently dated (e.g.  $^{210}Pb$ ), may be used to reconstruct ocean surface CO<sub>2</sub> and air-sea CO<sub>2</sub> fluxes over the past 200 years/the last glacial cycles and thereby provide crucial insight into ocean-atmosphere carbon fluxes since the onset of anthropogenic carbon emissions over deepwater formation regions. The application of the  $\delta^{11}B$  calibration to recent marine archives may therefore allow us to answer critical open

questions about the role of the Nordic Seas as a carbon sink when deepwater formation was reduced (stronger) and sea ice extent was greater (reduced).

## Data and materials availability

All data needed to evaluate the conclusions in the paper are presented in the paper and/or the Supplementary Materials.

## **Author contributions**


E.D.L.V. carried out palaeoceanographic sample preparation and geochemical analysis of plankton tows and core tops from expedition CE20009, compiled and analysed all datasets, and wrote the original draft of the manuscript. 710 A.M. conceptualized the project, acquired funding, carried out palaeoceanographic sample preparation of plankton tows from the Labrador Sea, coordinated the input of all co-authors and finalized the manuscript for publication. M.R. performed trace element and boron isotope analysis on N. pachyderma plankton tow samples from the Labrador Sea. J.B. supervised A.M. at the Alfred Wegner Institute and oversaw trace element and boron isotope analysis. U.N. oversaw all stable isotope analyses of water and foraminifera samples from CE20009 and 715 contributed to the final version of this manuscript. G.F. oversaw trace element and boron isotope analysis on N. pachyderma plankton tow samples from CE20009 and contributed to the final version of this manuscript. M.K. advised A.M. during project conceptualization, supervised her during her MSCA-IF at MARUM, University of Bremen, and contributed to the final version of this manuscript. T.B. provided support to E.D.L.V. during boron isotope analysis and contributed to the final version of this manuscript. J.C.B. carried out palaeoceanographic 720 sample preparation of plankton tows from expedition CE20009. M.E. provided sample material and contributed to the final version of this manuscript.

# **Competing interests**

The contact author has declared that neither they nor their co-authors have any competing interests.

## **Funding Sources**



This research was funded by MSCA-IF Project ARCTICO [838529] and the Marine Institute of Ireland Research Programme 2014-2020 (PDOC/19/05/02) awarded to AM. In addition, AM acknowledges the financial support of Science Foundation Ireland and the Geological Survey of Ireland under the SFI Frontiers for the Future Programme 21/FFP-P/10261 and Grant in Aid funding from the Marine Institute for research expedition CE20009 on the RV Celtic Explorer. M.R. acknowledges DFG (German Research Foundation) for funding through research grant no. RA 2068/4-1.

#### Acknowledgements

We gratefully acknowledge the support of the crew on the RV Celtic Explorer sailing under Master Anthony Hobin.

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
