# Peer review of "A $\delta^{11}B$ -pH calibration for the high-latitude foraminifera species Neogloboquadrina pachyderma and Neogloboquadrina incompta."

_EGUsphere, 2025_

## Author Comment (AC1)

**Response to Reviewer 1**

**Review #1**

**de la Vega et al:** A δ11B-pH calibration for the high-latitude foraminifera species Neogloboquadrina pachyderma and Neogloboquadrina incompta.

This study establishes a $\delta^{11}B$–pH calibration for two key high-latitude planktonic foraminifera species N pachyderma and N incompta. These species are critical for reconstructing past ocean pH and atmospheric $CO_2$ levels in Arctic and subpolar regions, where existing records are sparse. By using live-towed specimens collected from the North Atlantic, and pairing them with in situ hydrographic measurements, the authors avoid common assumptions required in core-top approaches and find:

1. Neogloboquadrina $\delta^{11}B$ values are lower than seawater $\delta^{11}B$ borate values. An interesting finding that is explained by localized acidification from respiration
2. The $\delta^{11}B$foram to $\delta^{11}B$borate slope is greater than 1, a notable deviation from other non-spinose species calibrations. Authors suggest that vital effects may be enhancing pH sensitivity in these species.
3. This calibration is applied to core-top samples from the Nordic Seas, demonstrating its paleoclimatic utility.

The motivation for the paper is sound and it is well-written. More studies like this are needed that explore the reasons why taxa can fall below the δ11B borate line and presumably are calcifying under pH conditions lower than ambient seawater. This work has implications for understanding boron isotope incorporation in other taxa with notably low δ11B (e.g. forams like H.elegans and bivalves). I would recommend the paper for publication following only minor edits. We thank Reviewer #1 for their positive review of our manuscript.

**Line by line points to consider:**

Line 43: Cite Rae et al., 2018 Nature too here. A boron isotope paper that demonstrates the importance of carbon cycling and climate in the Southern Ocean. We will add this reference in the revised manuscript.

Line 64: Degrees Celsius doesn't really need to be spelled out. Simply "°C" is unambiguous and widely understood. We will amend this as suggested in the revised manuscript.

Line 108: This paragraph implies that only tow data are used when also core top data appear in the methods. Signpost the reader here to say that the calibration in this study is only composed of the tow data (for the reasons mentioned), and it is compared to core top data to demonstrate paleoclimatic utility. We mention the use of coretops in the next paragraph line 125 "*In order to assess the validity of the calibration we construct, we then apply the tow-based calibration to a series of high latitude core tops alongside existing data from the literature to evaluate its application to the paleorecord*". We propose to move this sentence up to line 116 and thereby address the reviewer's comment.

Line 253: 0.5M HNO3 should be 0.5 M $HNO_3$ with a subscript. Check there is a space after M in other places where it is used in the text. We will amend this and check for consistency throughout in the revised manuscript.

Line 258: both Ca isotopes were measured, but presumably only 1 was used for calculating ratios? Each element ratios were calculated with $^{43}Ca$ and $^{48}Ca$ and then averaged. We will clarify this point in the revised manuscript.

Presumably Mg/Ca and Sr/Ca reproducibility are considerable better than 5%. We will clarify this in the revised manuscript and report the long-term reproducibility of the elemental ratios used in this study (e.g., Mg/Ca and Ba/Ca).

What [Ca] were samples run at? It says they were matrix matched, but to what concentration? 1mM? Ca concentrations ranged between 0.2 to 2 mM and were measured against a standard of the same concentration. To clarify we suggest rephrasing the sentence about Ca drift in lines 259-260 as follow: "*For samples bellow 1 mM, Element/Ca ratios typically drift, to account for this the concentration of the samples and standards were matched to account for this effect*".

Line 262: Stewart et al, 2020 Geostand Geoanal Res demonstrates the reproducibility of the Southampton method relative to other labs. This is important as the trace metal data in this study are from Southampton *and* AWI. It is important to demonstrate that consistent standard values were being hit by both labs so that the data can be compared directly without any interlab offset. We agree with the reviewer. In the revised manuscript we will report the long-term averages for Mg/Ca and Ba/Ca for NIST-C (reference standard) typically measured at Southampton (NOC) and show that they are consistent with values reported in Stewart et al. (2020). Similarly, we will report the long-term average for JCP-1 typically measured at AWI and compare these to the interlaboratory assessment published in Hawthorne et al (2013). In addition, we cite Stewart et al 2016 where JCP-1 values for Mg/Ca (measured at NOC) were reported. This comparison shows that both laboratories produce consistent values without any significant interlaboratory offsets. We will clarify these points in the revised manuscript and add Table 1 to the supplementary materials.

Table 1: Comparison of El/Ca standards measured at NOC and AWI

| Standards | NIST-C | | JCP-1 (uncleaned) | | |
|-----------|--------|-----|-------------------|-----|-----|
| El/Ca | Stewart et al. 2020 | NOC | Hawthorne et al. 2013 | AWI | NOC (Stewart et al. 2016) |
| Mg/Ca | 4.11±0.20 | 4.20±0.04 | 4.20±0.065 | 4.05±0.13 | 4.14±0.08 |
| Ba/Ca | 5.92±0.16 | 5.71±0.03 | 7.47±0.66 | 7.00±0.48 | n/a |

Line 266. New paragraph. We will add a new paragraph as suggested in the revised manuscript.

Line 267. 1.1 permil is a large TPB correction. It would be useful to know which sample this applied to in the data table with a column stating the magnitude of TPB correction. The Y axis error bars are large in Figure 2. It might be worth reiterating what went into these propagated errors (i.e. it is more that analytical uncert). This is important are there are two particularly low δ11Bforam values (~11.5 ‰) that it could be argued are driving the slope >1 phenomenon. It would be good to show clearly that these are not samples that are badly impacted by TPB. This is an important point. The low boron concentration of some samples resulted in a relatively large blank proportion (5% of sample size) and d11B correction of +0.4 to +1.1 ‰. For this reason, we aimed to be conservative and propagated the d11B uncertainty of the TPB (±2 ‰) into the corrected sample d11B, resulting in this large final uncertainty. These points are in the high d11B range of the calibration. The low d11B points are not TPB corrected and therefore give confidence in the final slope. Figure 1 and Table 2 below show that the calibration with and without the TPB-corrected samples both yield a slope >1. We propose to add the figure and table to the supplementary material and add a section following lines 368 where we outline that the TPB correction does not influence the slope >1 phenomenon.

[Figure]

Figure 1. Left: tow calibration including TPB corrected samples. Right: Tow calibration without TPB-corrected samples.

Table 1. The slope and intercept for each case is indicated below, with a slope > 1 in both.

|  | intercept | slope | mswd |
|---|---|---|---|
| **Tows (all sample)** | -11.09 (+/-5.91) | 1.58(+/-0.38) | 0.579 |
| **Tows (without TPB-corrected samples)** | -15.23(+/-7.31) | 1.84 (+/-0.47) | 0.700 |

Line 270: repetition of class 100. We will amend this in the revised manuscript.

Line 280: Again, worth citing Stewart et al., 2020 here as these are bang on interlab consensus values for AE121 that was found to be 19.63 ± 0.17 ‰. i.e. when measured it is quite a bit lower than the original Vogel value. We will cite Stewart et al. 2020 in the revised manuscript.

Also, I think you mean MINUS 20.2 ‰ for the AE120, right? For the JCp value Gutjahr et al., 2020 should also be cited here. Thank for you spotting this typo, we will amend this in the revised manuscript.

Line 312: As above. The use of JCp-1 here can be used to discount interlab offsets in Mg/Ca between labs. Was the JCp cleaned or unclean? How does this value 3.94 compare to Hathorne et al., 2013 and also the measurements made in Southampton? JCP-1 was not oxidatively cleaned at AWI. We will clarify this point in the revised manuscript. We will also revise the long-term average for Mg/Ca measured in JCP-1 (uncleaned, AWI) from 3.94 to 4.05±0.13 which is based on a more comprehensive dataset. We will then compare it to Hathorne et al (2013) and measurements made at NOC in Southampton (e.g., Stewart et al. 2016) in a supplementary Table (e.g., Table 1 above). This comparison will show that there are no significant interlaboratory offsets between AWI and NOC at Southampton for Mg/Ca.

Line 320: add these core location to the map figure with labels. We will add the core locations to the map in the revised manuscript

Line 325: GLODAP should be all capital letters. We will amend this in the revised manuscript

Line 327: modern pH/δ11Bborate is ambiguous (also line 368 – please check for other instances of this e.g. line 532). I think you mean "or" here, but it could be construed as the ratio of pH to δ11B borate. We will amend the sentence to "*unlike comparing foraminiferal $\delta^{11}B$ core top values to modern pH and $\delta^{11}B_{borate}$*".

Line 341: 432.6 μmol kg−1 needs a superscript. Otherwise it just means minus 1. We will amend this in the revised manuscript

Line 346: Does this monte carlo approach need a citation of Chalk et al., 2017? This method was used by several boron studies including Martinez-Boti et al. (2015) and Chalk et al. (2017), this detail will be added to the revised manuscript.

Line 355: ok to group them for now, but it will be fascinating to see if N. pachyderma and N. incompta need to be separated for calibration purposes once more δ11B data are generated across a wide pH range for each taxon. The differences in Ba/Ca noted later in the text are intriguing and suggests they are living very different lives. I look forward to the next study on this!! Yes, we agree.

Line 388. I think this section 3.3 is what is needed in the intro to signpost the reader about what is to come before they see the methods. We have addressed this point above.

Line 410: Could even bring coral data in here for comparison to show that this slope >1 is really unusual. You already use the Balanophyllia Gagnon 2021 citation elsewhere. Could also mention D.dianthus from Anagnostou 2012 and McCulloch et al., 2012. We will add the suggested references in the revised manuscript

Line 423: again "δ11Bforam-δ11Bborate" is an ambiguous dash that could be a minus sign We will amend this (revise into a dash) in the revised manuscript

Line 456: Unsure what "privileged" is supposed to imply here. I'm not sure it is needed. We will delete the word "*privileged*" in the revised manuscript

Line 471: CO3-2 should be "2-" not to the power of minus 2. We will amend this mistake in the revised manuscript

Line 481: use words as part of a sentence rather than >> here. We will amend this sentence to: "*even if the pH internally is significantly greater than the micro-environment*", in the revised manuscript.

Line 520: pH written twice We will delete the repition in the revised manuscript

Line 565: space between units "50 m" and "150 m". There are many other instances of this in the text. Please check. We will add the space as suggested and check for consistency throughout the revised manuscript.

Line 566: d11Bborate is now italic. Be consistent for clarity. We ensure consistency throughout the revised manuscript.

Line 601: "is" shouldn't be italic We will correct the font in the revised manuscript

Line 672 "extent" We will amend this in the revised manuscript

Table 1: add the tpb correction magnitude to this table. Also check the number of significant figures on the Ba/Ca column. We will add the TPB correction to this table in the revised manuscript.

Table 2: define mswd. We will add "mean square weighted deviation" to this sentence in the revised manuscript

Figure 1: Caption says pCO2, but this is delta pCO2 relative to the atmosphere. This should be clearly explained in the caption. The figures could be better integrated with one another for instance plankton

tows are red dots in Fig1 and therefore should be red dots in the data figure 2. Try to keep colours and symbols consistent to guide the reader's eye of what is incompta and pachyderma and what is a tow or a core top. Use larger symbols for "this study" if you need to draw attention to what is new. We will streamline and amend this figure in the revised manuscript

Figure 2: axis labels are ambiguous. The use of a dash (−) in "Foraminifera (tow) – δ11Bforam (‰)" implies this is some value of foraminifera minus the $\delta^{11}B$ of calcite (same goes for the x axis). Y axis should read "$\delta^{11}B_{foram}$ (tow) [‰]" and x axis should read "Seawater $\delta^{11}B_{borate}$ [‰]". We will clarify the axis label in the revised manuscript.

Figure 3: more detail needed in the caption. We will add more detail in the figure caption and explain that it is a model.

Figure 4: again check for ambiguous minus signs. Do you actually mean minus in this case of d11Bforam – δ11B borate? It is not clear which taxa are plotted in panel b. the caption says pachyderma, but the extremely high Ba/Ca values suggest incompta are also on there. As per my earlier comment – figures need to be better integrated with the same colours and or symbols for tows and species where possible to guide the reader's eye. We will clarify the axis label and colours used for this figure in the revised manuscript.

Figure 5: as above. Axis labels need revision for clarity. Caption needs to explain again PI and $C_{ANT}$ so that the reader doesn't need to search the main text. We will add "*PI stands for preindustrial and $C_{ANT}$ for anthropogenic carbon correction.*" in the figure caption in the revised manuscript.

Figure 6: y axis should read "…concentration [ppm]". d18Oc should be $\delta^{18}O_c$. N pachyderma should be italic. Explain "CIAAN" Station. Again this could be better integrated with the map figure. e.g. Station 16 is mentioned often in the text but not highlighted on the map. We will amend the y-axis as suggested and clarify the figure caption in the revised manuscript. We feel that station IDs on the map would lead to overcrowding. Detailed locations (Lat/Long) for each station are listed in Supplementary Table S1.

---

## Author Comment (AC2)

**Response to Reviewer 2**

**Review #2**

De la Vega et al presented a new calibration for boron isotopes in polar ocean planktic foraminifera Neogloboquadrina pachyderma/incompta using specimens from plankton tows. The authors explored why the slope of the calibration between d11B of foram and borate is larger than 1 and used new and published core top data to validate the new calibration.

The manuscript is well-written, and the new calibration is valuable to future paleo pH/CO2 reconstructions in the high-latitude oceans. We thank Reviewer #2 for their positive review of our manuscript.

The only major suggestion I have is to consider recalculating downcore pH/pCO2 from d11B reported in Yu et al 2013 and use this to further validate the new calibration. This is a valuable comment. However, applying our calibration to the downcore data of Yu et al (2013) is the objective of a subsequent manuscript that includes new downcore d11B data from the Nordic Seas. It will make more sense to recalculate Yu's data in the context of this forthcoming study, integrating pH/CO2 data on a regional North Atlantic scale. Consequently, we prefer not to include this recalculated data that would deserve an entire palaeoceanography section, context, and interpretation and we aim here to focus solely on the calibration.

However, to acknowledge the reviewer's point, we propose to add the following sentence to line 639. "*Applying the new calibration with a slope >1 to the downcore data of Yu et al. (2013), will elevate reconstructed surface seawater CO2. As a result, regions off Iceland in the polar North Atlantic may not have served as a sink of $CO_2$ to the atmosphere during the glacial and deglacial period as hypothesized in Yu et al. (2013). A detailed analysis of existing and new palaeoceanographic datasets will be the object of a forthcoming manuscript*".

**Minor comments**

L113: While I appreciate the advantage of a calibration based on plankton tows, I think it would be great to mention the caveat that the depths of the plankton tows are not necessarily the depth of calcification depths of the forams. We will add the following sentence to the revised manuscript: "*Furthermore, we acknowledge that the depth of the plankton tows does not necessarily represent the depth of calcification.*"

Fig 1: DpCO2 is shown in this figure. Calculated d11B of borate or pH would be better for the map. We show delta PCO2 here because it is valuable for the application of the calibration to modern core tops and the calculation of CO2 in section 4.3.

Table 1: It would be great to have TA and DIC data in this table or in a supplementary table We will add TA and DIC in a supplementary table in the revised manuscript.

Show raw and TPB-corrected d11B for all samples. Maybe also add a supplementary figure showing the calibration with raw d11B data to demonstrate that the large TPB correction (for some samples) is not driving the slope of the calibration. The figure and table below (e.g., Figure 1, Table 1) shows the calibration with all samples without a TPB correction (left, samples with strong TPB contamination are given a 2‰ uncertainty to account for this lower confidence) and with TPB-corrected samples removed (right). In both cases the slope remains >1. We are happy to follow the reviewer's suggestion and will add the figures and table to the supplementary material. We will also add a section following lines 368 where we outline that the TPB correction does not influence the slope >1 phenomenon.

[Figure]

Figure 1: Left: tow calibration without the TPB correction. Right: Tow calibration without TPB-corrected samples.

Table 1: The slope and intercept for each case is indicated below, with a slope > 1 in both.

|  | intercept | slope | mswd |
|---|---|---|---|
| Tows (all sample with TPB correction), this study | -11.09 (+/-5.91) | 1.58(+/-0.38) | 0.579 |
| Tows (with TPB-corrected samples removed), right figure. | -15.23 (+/-7.31) | 1.84 (+/-0.47) | 0.700 |
| Tows (all sample, no TPB correction), left figure. | -9.07 (+/-5.88) | 1.45(+/-0.38) | 0.663 |

L202-241: This part is mostly about deriving hydrographic data to calculate borate d11B. Consider moving these before L191 where the calculation of borate d11B was introduced. We will do as suggested in the revised manuscript.

L205: The logic of constraining the calcification depth is not clearly described. I recommend adding a sentence stating that the calcification depth is determined by "comparing the d18Oc with equilibrium d18O in the water column". We will do as suggested in the revised manuscript.

L210: N. pachyderma d18O, along with other data to decide calcification depth needs to be presented in a supplementary table. We will provide this data in a supplementary table as suggested in the revised manuscript.

L261: new paragraph before "Samples for boron..." We will amend this in the revised manuscript
L261: "boron [isotope] analysis"

L272: how the sigma is calculated is confusing. If the Anagnostou equation is employed, the sigma can be derived from 11B voltage and does not need to be derived from JCP-1 measurements. This is correct, the 11B voltage of each sample (not JCP-1) is used to determine the uncertainty. We will clarify this in the text as follows. "*The uncertainty is dependent on the boron content (Rae et al., 2011), i.e. the intensity of the $^{11}B$ signal of each sample*".

L278: Some reorganization is needed here. It is essentially an interlab measurement comparison, but AWI measurements have not been introduced at this point. Yes, that is correct. We will add a sentence at the beginning of section 2.3 to clarify this point in the revised manuscript

L280: "-"missing for d11B of AE120. We will amend this in the revised manuscript.

L302: Need to be reorganized. Triplicated measurements were mentioned before being introduced in L305. Yes, we agree. We will move the sentence from line 302 down one sentence to clarify this point.

L311: Unclear what 0.95+-0.47% is. It reads like this is the rsd for multiple El/Ca ratios, which would not be a proper way to report analytical errors. Yes, the reviewer is correct, these values refer to the RSDs reported in % associated with replicate analysis of the same sample. As part of the revisions, we will report the long-term average values of reference material (e.g., JCP-1 and NIST) measured at both AWI and Southampton labs also summarized in the table below. This comparison will also show that there are no significant interlaboratory offsets between AWI and NOC at Southampton for Mg/Ca.

Table 1: Comparison of El/Ca standards measured at NOC and AWI

| Standards | NIST-C | | JCP-1 (uncleaned) | | |
|---|---|---|---|---|---|
| El/Ca | Stewart et al. 2020 | NOC | Hawthorne et al. 2013 | AWI | NOC (Stewart et al. 2016) |
| Mg/Ca | 4.11±0.20 | 4.20±0.04 | 4.20±0.065 | 4.05±0.13 | 4.14±0.08 |
| Ba/Ca | 5.92±0.16 | 5.71±0.03 | 7.47±0.66 | 7.00±0.48 | n/a |

L355: i.e. not e.g. We will amend this in the revised manuscript.

L360: including the number of data points will be good. We will add the number of samples (e.g., n=16) in the revised manuscript)

L364-367: unclear as written. Suggest rephrasing. We will clarify this sentence in the revised manuscript to: *"Furthermore, the slope lies below the 1:1 δ11Bforam: δ11Bborate line. When assessing N. pachyderma samples by themselves (i.e. tows from the Labrador Sea) the slope is steeper at 1.82. However, both slopes and intercepts are within error of each other (Table 2)."*

Fig 2: explain what errorbars are in panel 1 and what blue shades are in panels c,d. We will clarify these points in the revised figure caption.

X axes for panels A,B: borate d11B or d11B_borate. We will change the axis as suggested in the revised manuscript.

Y axis for panel B: foram d11B or d11B_foram. We will change the axis as suggested in the revised manuscript

L383: there's no fig. 2e. Thank you for spotting this typo. We will correctly refer to Fig S1 in the revised version of this manuscript.

L449: not sure how applicable the Hönisch 2019 method is here. If ALK and DIC are lowered by 400 and 200 umol/kg, diffusion will elevate these concentrations. Maybe. Yes, the model proposed by Hönisch 2019 is a simple model ignoring any subsequent effects and processes such as diffusion. However, we feel that this thought experiment provides a valuable point in our discussion. We are happy to caveat this point in the revised version of this manuscript.

L481: replace ">>" We will amend this in the revised manuscript.

L620: Sentence not complete. Thank you for spotting this. We will remove the paragraph break that is separating this sentence from its end in line 621.

---

## Author Response (AR2)

**Author's Response to the Associate editor and Reviewers**

**Associate editor:** The manuscript was evaluated once again by two experts in field. The reviewers and I agree that the authors overall did a good job addressing the comments provided to them during the first rounds of reviews. However, both reviewers provided some additional suggestions to improve the clarity of the text, in particular with respect to the Materials and Methods used. I kindly ask the authors to address these minor revisions. Once they do so, the manuscript will be ready to be accepted for publication in Biogeosciences.

Thank you for your positive assessment, we are happy to make these last changes. As suggested below

**Reviewer 1:** The authors have generally done a good job of addressing the previous comments. I would recommend the manuscript for publication pending correction of a few very minor errors.

• The response to review states that Rae et al., 2018 has been added to the Ms, but as far as I can tell, it hasn't. If the authors have elected not to add this reference, ok, but please be accurate in the response to review.

We are very sorry for this oversight. We are happy to include the reference as suggested in the first review. It is now inserted in 11.43 of the revised manuscript

• Fig caption 1. The plankton tow sites have been added – great. However, the caption says "red circles" when the points are black. "pCO2" needs a subscript both times it has been added here.

The figure caption has been revised as suggested

- Table 1 and methods:
- o Please refer to "NIST RM 8301 (Coral)" by its full name rather than just the ambiguous "NIST-C".

  Done as suggested
- o The table says Stewart 2020. This should be 2021 I think.

It is cited as Stewart et al 2021, and yes that is the correct reference

o Please also be consistent in reference to the "Southampton" lab. When the text says samples were measuring in "Southampton" and the table says "NOC" this is unclear to the reader. Best to use Southampton throughout. Perhaps also add "this study" to the relevant box in the table 1 as there are two examples of Southampton reproducibility.

Done as suggested

o What are the  $\pm$  values in table 1? This isn't mentioned in the caption. Are they 2 sigma or 1 sigma? Are they SE or SD? I am not sure like-for-like is being compared here, particularly when inter-lab consensus values are being compared to individual lab reproducibility. Please be clear on what is being presented.

The reviewer is correct. In the revised Table 1 we have specified exactly what the uncertainty refers to (e.g., SD, Robust SD and Residual SD) as it was originally reported.

• Table 3. Please add somewhere that these slope and intercept values are "linear" calibration fits to the data (i.e. in the form y=mx+c)

We have added this information as requested in the Table title. Specifically, we added: Slopes and intercepts were fitted with a york linear regressions of the type: y=mx+c.

• The number of significant figures in the reported Ba/Ca values in Table 2 have not been addressed. For example, if quoting 3 sig figs, then the value 601.80 should just be  $602 \ \mu mol/mol$ .

We have amended the table and presented all Ba/Ca data with 3 significant figures.

**Reviewer 2:**

The authors addressed most of my comments from my last review. I am happy to recommend acceptance of the manuscript with some minor revisions to the method section.

**Detailed comments** Section 2.3 I acknowledge that the analytical methods of this work are a bit difficult to summarize, as it was done in two different labs. Some of the methods are repeated (e.g., d11B measurements); some can be moved to somewhere else for a better flow (e.g. 274-279). I suggest the authors reorganize and shorten this section for better readability.

Since the methods are quite different between the two laboratories, we were only able to make moderate changes. But we have shortened the text when possible (line 323-325) to avoid repetition during sections 2.3.1 and 2.3.2

L266: break this sentence into 2.

Done as suggested.

L269: not sure what bracketing standards are.

We have corrected this typo to "consistency standards" and amended the text to "consistency standards in-house standards placed at the beginning and end of each sequence were measured at the same concentration as samples to assess accuracy." We have clarified this in the revised manuscript.

L269 "sample below 1mM" is not clearly written

We have amended the text to "1 mmol/mol".

L295: My previous comments on the propagation of the uncertainties in d11B were not addressed. The description is contradictory to itself as currently presented.

The JCP-1 measurements were only used to define the relationship between d11B uncertainty and 11B signal intensity (published in Anagnastou et al., 2019). We have rearranged the text as follow to clarify this point.

"The uncertainty on foraminifera  $\delta^{11}B$  is dependent on the boron content (Rae et al., 2011), i.e. the intensity of the  $^{11}B$  signal of each sample. The relationship between  $^{11}B$  signal and sample uncertainty was empirically determined based on the uncertainty of repeated measurements of JCp-1 Coral (Porites sp.) that has undergone the same chemical purification It is defined by the following equation (Anagnostou et al., 2019)".

L 410: add "respectively" after uncertainty

Done as suggested

L488: caveat is not a verb

We have replaced "caveat" with "note"

L683: add "strong" before "sink". Nuanced writing would be helpful if you are not presenting detailed arguments against the previous study.

Done as suggested